# Bread Products from Blends of African Climate Resilient Crops: Baking Quality, Sensory Profile and Consumers’ Perception

**DOI:** 10.3390/foods12040689

**Published:** 2023-02-05

**Authors:** Stefano Renzetti, Heikki Aisala, Ruth T. Ngadze, Anita R. Linnemann, Martijn W. Noort

**Affiliations:** 1Wageningen Food and Biobased Research, Wageningen University & Research, Bornse Weilanden 9, 6700 AA Wageningen, The Netherlands; 2VTT Technical Research Centre of Finland Ltd., Tietotie 2, 02044 Espoo, Finland; 3Food Quality and Design Group, Wageningen University and Research, Bornse Weilanden 9, 6708 WG Wageningen, The Netherlands

**Keywords:** bread, sorghum, cowpea, cassava, sensory, food security

## Abstract

With food insecurity rising dramatically in Sub-Saharan Africa, promoting the use of sorghum, cowpea and cassava flours in staple food such as bread may reduce wheat imports and stimulate the local economy through new value chains. However, studies addressing the technological functionality of blends of these crops and the sensory properties of the obtained breads are scarce. In this study, cowpea varieties (i.e., Glenda and Bechuana), dry-heating of cowpea flour and cowpea to sorghum ratio were studied for their effects on the physical and sensory properties of breads made from flour blends. Increasing cowpea Glenda flour addition from 9 to 27% (in place of sorghum) significantly improved bread specific volume and crumb texture in terms of instrumental hardness and cohesiveness. These improvements were explained by higher water binding, starch gelatinization temperatures and starch granule integrity during pasting of cowpea compared to sorghum and cassava. Differences in physicochemical properties among cowpea flours did not significantly affect bread properties and texture sensory attributes. However, cowpea variety and dry-heating significantly affected flavour attributes (i.e., beany, yeasty and ryebread). Consumer tests indicated that composite breads could be significantly distinguished for most of the sensory attributes compared to commercial wholemeal wheat bread. Nevertheless, the majority of consumers scored the composite breads from neutral to positive with regard to liking. Using these composite doughs, chapati were produced in Uganda by street vendors and tin breads by local bakeries, demonstrating the practical relevance of the study and the potential impact for the local situation. Overall, this study shows that sorghum, cowpea and cassava flour blends can be used for commercial bread-type applications instead of wheat in Sub-Saharan Africa.

## 1. Introduction

Currently, the food system in Sub-Saharan Africa (SSA) is severely challenged because of the growing urban population and climate change [1,2]. Massive urbanization is resulting in a dietary shift, diverting from the use of local crops to imported cereals associated with the Western lifestyle, largely maize and wheat [3]. Concomitantly, food production cannot keep pace with internal demand, making SSA increasingly dependent on imported crops such as wheat [4]. This issue is further exacerbated by climate change with projected reductions in wheat and maize yields in SSA [3,5] and by the wheat crisis related to the Russia–Ukraine conflict.

Indigenous African crops such as sorghum, cassava and cowpea are climate-resilient crops (CRCs) as they have enhanced tolerance to biotic and abiotic stresses [6,7]. Cowpea requires few inputs, grows in low rainfall and shade [8] and is suitable for intercropping with sorghum, reducing fertiliser use and soil erosion [9]. Promoting the valorisation of these CRCs into value-added ingredients and in the production of attractive, affordable and nutritious food such as bread-type products could improve food and nutrition security, stimulate the local economy and create new employment opportunities throughout the value chain [10,11].

From a nutritional perspective, the relative overconsumption of starchy crops in SSA is a concern for protein as well as micronutrient security [12]. Cassava is an affordable source of carbohydrates and energy, but should be combined with other indigenous crops from pulses and cereals [13]. Consumption of sorghum can positively impact human health, especially regarding disorders such as celiac disease, diabetes and obesity [14]. Tannins present in sorghum can reduce starch digestion [15,16], resulting in a low glycaemic index [17,18]. Monomeric flavonoids in sorghum showed several anti-inflammatory activities [19]. Sorghum is also rich in potassium and phosphorus and contains good levels of calcium. The legume cowpea is a rich source of protein, dietary fibre and additionally of polyphenols, vitamins and minerals [20,21]. Cowpea proteins are rich in glutamic acid, aspartic acid and lysine, which are limiting in sorghum. From this perspective, it is nutritionally beneficial to complement sorghum and cassava with cowpea in the human diet [22].

Sorghum flour has been studied for partial wheat replacement in pan breads [23], showing detrimental effects on dough properties and the quality of the composite breads with inclusions of 15% and beyond. From a sensory perspective, the inclusion of sorghum in wheat bread products was found to be acceptable up to a maximum of 30% [24,25]. Due to its gluten-free status and nutritional value, sorghum flour has been also studied for application in breads for people with celiac disease [26], often in combination with starches [26,27] and with hydrocolloids [28]. Cassava has been extensively studied for partial replacement of wheat flour in bread applications [29]. However, widespread implementation of cassava in commercial products has failed to date, due to technological challenges and poor product quality of cassava-wheat composite breads as perceived by consumers and bakers [11].

Despite the relevance for human nutrition and food security, studies on composite breads based on sorghum, cowpea and cassava are scarce. However, their combination offers opportunities to optimize the nutritional composition and technological properties of bread-type products [30], which would benefit the growing African population but potentially also the segment of the global population suffering from celiac disease. To the best of our knowledge, Renzetti and co-workers were the first to report on the use of a composite sorghum–cowpea–cassava mixture to prepare a processable dough, instead of a liquid batter, and produce tin breads (i.e., breads baked inside tins) [31]. These authors reported that (physical) texture properties of the breads could be improved by modulating cowpea flour properties, e.g., by dry-heating treatments, suggesting cowpea as a functional ingredient for both technological and nutritional purposes. Further studies were suggested to explore the implementation potential of these bread concepts, taking also the sensory aspects into account. Variations in cowpea varieties and inclusion levels may have a different contribution to the quality of the bread-type products due to differences in sensory profile [32] and in functional properties [33].

Against this background, this study evaluated the effect of variations in cowpea flour properties, namely variety and dry-heating treatment, and the level of addition (i.e., cowpea to sorghum ratio) on the physical and sensory properties of breads made from sorghum, cowpea and cassava mixtures. Two cowpea varieties were used for this purpose. Relevant physicochemical properties of the flours were analysed in order to relate to the bread properties. Descriptive sensory analysis was performed to understand the effect of variations in flours and bread properties on the intensity of the sensory attributes. Additionally, consumer studies were performed to gain insights into the perception and liking of a selected CRCs-based bread formulation as compared to commercial wheat bread. Implementation potential of the developed doughs was tested in Uganda with small bakeries making tin breads and with street vendors making chapati.

## 2. Materials and Methods

### 2.1. Materials

Red, non-tannin, King Korn superfine sorghum (locally known as mabele) meal was obtained from a local supermarket in Pretoria (South Africa). Cowpea seeds from a white (Bechuana white) and a red (Glenda) variety were sourced from Agrinawar Agricol Pty (Pretoria, South Africa) and milled as whole grains at ProCorn Mills (Edenvale, Sebenza, South Africa). Sorghum contained 10.2% protein, 11.5% dietary fibres (of which 2.3% was soluble) and 73.1% starch. Bechuana white cowpea flour (BECH) contained 23.9% protein (of which 18.6% was soluble at native pH), 20.6% dietary fibres (of which 7.5% was soluble) and 40.7% starch. Red Glenda cowpea flour (GL) contained 23.9% protein (of which 18.4% was soluble at native pH), 24.1% dietary fibres (of which 7.7% was soluble) and 39.7% starch. Dry-heated GL (GL-DH) was also tested based on the result of a recent study on the Glenda variety [31]. One treatment condition was selected to evaluate the contribution to the sensory profile, which was not previously assessed. The treatment consisted of equilibrating the flour at 50% RH and subsequently heating in an oven at 100 °C for 2 h after sealing in a heat-resistant plastic bag.

Cassava starch (93.2% starch, 0.3% protein and 2.1% soluble fibres) was supplied by DADTCO (Dutch Agricultural Development and Trading Company, Inhambane, Mozambique). Psyllium husk powder (88% dietary fibres, 3% proteins) was from Unilecithin (Sharjah, United Arab Emirates). Dry yeast (Mauripan red, AB Mauri, Dordrecht, The Netherlands), salt and sucrose (both sourced in The Netherlands for lab tests and in Uganda for implementation tests) were used for the breadmaking trials.

### 2.2. Methods

#### 2.2.1. Water Binding Capacity of Native and Treated Flours

The water binding capacity (WBC) of the native and dry-heated GL (GL-DH) samples was determined in triplicate, according to a modified version of [34]. Flours (0.4 g on dry basis) were placed in 5 mL Eppendorf tubes and 3.6 g of distilled water was added during vigorous stirring. After mixing on a vortex, the samples were left to shake at room temperature for 20 min on a Multi Reax Vortex (Heidolph Instruments GmbH, Schwabach, Germany). Then, the samples were centrifuged for 10 min at 5000 G using an Avanti J-26XP High Speed Centrifuge (Beckman Coulter, Indianapolis, IN, USA). The supernatant was collected and the pellet was drained for 15 min at an angle of 45°, dried and then weighed. WBC was expressed as follows:(1)$$WBC\;\left(g/g\right)=\frac{\mathrm{wet}\;\mathrm{pellet}\;\left(g\right)-\mathrm{dried}\;\mathrm{pellet}\;\left(g\right)}{\mathrm{dried}\;\mathrm{pellet}\;\left(g\right)\;}$$

The moisture content of the pellet was measured by drying overnight in aluminium dishes in an oven at 105 °C. The filled dishes were cooled for 1 h in a desiccator before weight determination.

#### 2.2.2. Thermal Analysis of Native and Treated Flours

Flour concentrations of 20% (on dry matter basis) in distilled water were used to measure starch gelatinization and protein denaturation with a TA Instruments type Q200 Differential Scanning Calorimeter (DSC) as earlier described [31]. Samples were measured by equilibrating at −5 °C for 5 min and then scanned to 160 °C with a rate of 5 °C/min. The onset of starch gelatinization and protein denaturation (T_onset_), peak temperature (T_min_) and gelatinization/denaturation enthalpy were determined using the analysis tool available in the Universal Analysis software (TA Instruments, New Castle, DE, USA). Experiments were performed in triplicate.

#### 2.2.3. Pasting Behaviour of Native and Treated Flours

Pasting behaviour was investigated using a Rapid Visco Analyser (RVA) Super 4 (Perten, Hägersten, Sweden) as earlier described [31]. Flour or starch suspensions of 8% dry matter (dm) in water were used. Samples were subjected to a time–temperature profile. Initial stirring speed was 960 rpm at 50 °C for 60 s. Then, the stirring speed was decreased to 160 rpm while the temperature was increased to 95 °C within 3 min 42 s. Samples were then held at 95 °C for 2 min 30 s minutes and cooled to 50 °C within 3 min 48 s. Finally, samples were held at 50 °C for 2 min. All tests were performed in duplicate. Pasting temperature (PT), peak viscosity (PV), hold viscosity (HV), breakdown (BD), final viscosity (FV) and set back (SB) were obtained from the RVA measurements and expressed as centipoise (cP).

#### 2.2.4. Bread-Making Procedure

Variations in CRCs-based bread formulations were built upon a product concept recently developed containing sorghum, cassava and cowpea [31], as shown in Table 1. Variations were designed to test the effect of cowpea flour (i.e., GL9 vs. G27), compare between varieties at highest inclusion level to maximize differences (i.e., G27 vs. BENCH27) and evaluate the effect of dry-heating treatment as compared to untreated cowpea flour (i.e., GL9 vs. GL9-DH).

In total, about 1300 g of dry ingredients were added to a Sinmag spiral mixer (Sinmag Europe bvba, Zuienkerke, Belgium) and pre-mixed at low speed (speed I) for 1 min. Then, water was slowly added during mixing, which was performed for 5 min at speed I and for another 4 min at speed II. After mixing, the dough was divided and shaped manually, and put into three greased baking tins, each containing 700 g of dough. These tins were put in a fermentation chamber at 30 °C and 85% RH. The proofing time was defined as the time needed by 50 g of dough to reach a CO_2_ production of 90 mL. The CO_2_ production was determined using a Risograph (National Manufacturing, Lincoln, NE, USA). After proofing, the doughs were put in a swing oven at 180 °C for 70 min. During the first minute, steam was injected twice to regulate the moisture content. After baking, the breads were cooled at room temperature for 40 min, sealed in plastic low-density polyethylene bags and stored at room temperature until further analysis one day after baking. Baking tests were performed in triplicate. The described procedure was used for bread intended for instrumental characterization and descriptive sensory analysis.

For the naïve consumers’ study, sorghum and BECH flours were first hydrated in excess water for 1.5 h in order to minimize sandiness in breadcrumbs, based on preliminary trials. After hydration, the flour and the water were combined with the rest of the ingredients in the Sinmag spiral mixer and the same breadmaking protocol was applied.

#### 2.2.5. Instrumental Bread Quality Evaluation

Loaf volume was determined on 2 loaves from each baking test, with a rapeseed displacement according to [35]. In total, 6 measurements were performed per variation. Specific volume (SV) was calculated as loaf volume divided by loaf weight (mL/g).

Crumb texture was measured by means of Texture Profile Analysis (TPA), using a TA-XT2i Texture Analyser from Stable Micro Systems (Godalming, Surrey, UK) with a 30 kg load cell and a 75 mm compression plate and performed as previously described [35]. In total, 12 measurements were performed per bread type.

The moisture content of the bread crumb (5 g sample) was measured according to [35] by drying overnight in aluminium dishes in an oven at 105 °C. The filled dishes were cooled for 1 h in a desiccator before weight determination. In total, 6 measurements were performed per bread type.

#### 2.2.6. Descriptive Sensory Analysis

Sensory profiling of the pan breads was conducted as generic descriptive analysis with 8 trained panellists who were regularly trained. Written informed consents were obtained from the participants prior to the evaluation. The base attribute list was developed by five trained assessors in a consensus session using previous studies as a basis [32,36,37]. This list, along with all samples, was then presented to the whole sensory panel in a consensus training session where they refined the attributes and their descriptions, discussed the intensity ranges of the samples and decided on reference products. The final sensory profile had 18 attributes that covered the odour (i.e., cereal, sweet, ryebread, beany, fermented), appearance (dark, air bubble size), tactile texture (crumbliness), taste (sour, sweet, astringent), flavour (ryebread, beany, yeasty, spicy) and mouthfeel (sandy, crumbly, moist) of the samples.

For serving, a slice of each sample was packed in closed plastic bag and presented with a 3-digit code. Water and a piece of wheat-based cream cracker were used as palate cleansers. The serving order was randomized with a Latin squares design. The samples were evaluated in duplicate in a complete block design. The attribute intensities were evaluated with a 0–10 continuous line (0 = non-perceivable and 10 = very intense). The data were collected with EyeQuestion version 5.3 (Logic8 B.V., Elst, The Netherlands).

#### 2.2.7. Sensory Evaluation with Naive Consumers of a CRCs-Based Bread and of Wholemeal Wheat Bread

##### Subjects

Fifty-one participants (*n* = 51, 37 female, age: 19–28 years) were recruited from Wageningen University & Research campus using flyers, posters and social media. Subjects were regular consumers of bread. Other inclusion criteria (self-reported) were no allergies or intolerances to gluten, good dental health and non-smoking habits. Participants were mainly of Dutch nationality (*n* = 37), followed by Italian (*n* = 5), and the remaining 9 were each from a different nationality. A consent form was signed by all participants. Subjects received reimbursement for their participation and were naive concerning the experimental conditions and purposes.

##### Sensory Sessions

The sensory tests were conducted in meeting facilities at Wageningen University, equipped with desk dividers for a maximum of six participants per session. Subjects were asked to fill in a paper questionnaire. One short session was carried out to allow participants to familiarize themselves with the sensory method. An explanation brochure for the different descriptors and their definitions was provided during the test session. Mechanically sliced samples of bread (thickness 1 cm) of the two different formulations were presented to the panellists with randomized three-digit codes. The panellists received one slice of bread per formulation. Panellists were instructed to first taste the bread crumb of the two different bread samples before scoring them. Subjects were instructed to cleanse their mouth with water and have a break of at least 2 min between evaluating the samples.

##### Hedonic Characterization and Rate-All-That-Apply (RATA)

Participants were asked to evaluate commercial wholemeal bread (Stevig grof volkoren, Albert Heijn, Zaandam, The Netherlands) and CRCs-based bread prepared following the recipe of GL9 (Table 1), but using BECH instead of GL. Participants first evaluated overall liking of the bread using a hedonic 9-point scale ranging from “Dislike it extremely” (1) to “Like it extremely” (9). After the hedonic evaluations, subjects were asked to evaluate the samples using a RATA method with 9-box scales as previously described by [38,39]. The complete list of sensory terms, as well as their definitions, are reported in Table 2. The list of attributes was presented to the subjects, who were asked to indicate whether the specific descriptors were applicable to the assessed sample (“Yes” or “No” choice). Once an attribute was selected as applicable to the sample (“Yes” choice), then subjects had to rate the perceived intensity of the selected attribute on a 9 point-scale where “1” corresponded to low intensity and “9” to high intensity. It was clarified that a non-selection of an attribute was equivalent to a non-perception of the sensory stimulus. The order in the questionnaire of the sensory attributes was randomized within each block of attribute category (appearance, texture, and flavour) for each participant.

#### 2.2.8. Statistical Analysis

Statistical evaluations (analysis of variance, ANOVA, with Tukey’s test as post hoc test at a significance level of *p* < 0.05) for flour properties and physical properties of breads were performed with SPSS (IBM, version 25, Chicago, IL, USA). A two-way mixed model ANOVA with samples as fixed factor and assessors as random factor was used for sensory profiling. Principal component analysis (PCA) of sensory profiling data was performed with Rstudio (RStudio version 1.1.463, Inc., Boston, MA, USA) using the PCA function of the *FactoMineR* package [40], together with correlation analysis. Averaged data over assessors and replicates were used for the PCA on sensory profiling.

For the RATA intensity scores, non-checked attributes were treated as intensity = 0, and RATA intensity scores (0–9) were treated as continuous data [38,39]. A paired sample *t*-test was performed to establish significant differences between the two bread samples for each of the listed attributes. A significance level of *p* < 0.05 was chosen. Data from RATA tests were analysed using SPSS.

## 3. Results and Discussion

### 3.1. Physicochemical Properties of Sorghum, Cowpea and Cassava Flours

Relevant physicochemical properties of the flours were studied to elucidate their contribution to the baking behaviour of the composite CRCs doughs (Table 3). GL showed the largest WBC among all flours, followed by BECH, sorghum flour and cassava starch. The WBC of dry-heated GL flour was intermediate to GL and BECH. The decrease in WBC of cowpea flour with dry-heating may be attributed to increased hydrophobicity of proteins and annealing of starch [31,41]. The main compositional difference between the two cowpea varieties was in the total dietary fibre content and particularly the insoluble part. Most likely, these compositional differences explained the WBC results, as a general correlation was observed between WBC and the dietary fibre content of the flours (R^2^ = 0.968, *p* < 0.05).

The DSC analysis of the flours revealed the main differences as being in their thermal behaviour (Table 3). Cassava and sorghum flour showed one main endothermic peak associated with starch gelatinization. The gelatinization temperature for cassava and sorghum starch were in agreement with ranges previously reported [42,43]. On the contrary, GL and BECH flours were characterized by two endothermic transitions with peaks appearing around 77 and 88 °C. These peaks could be associated with starch gelatinization and protein denaturation, respectively, as earlier reported [31]. Cassava showed the lowest onset of starch gelatinization (T_onset_), while sorghum and BECH had the highest (*p* < 0.05). The T_peak_ of starch was the highest for BECH, followed by GL and GL-DH, while cassava was the lowest (*p* < 0.05). The enthalpies for starch and protein in cowpea flours could not be distinguished due to partial overlapping between the two peaks. Nevertheless, the gelatinization enthalpies of cassava and sorghum flour were significantly higher than the enthalpy for all the cowpea flours.

With regards to pasting behaviour, cassava starch showed the lowest pasting temperature while sorghum flour had the highest, followed by BECH (*p* < 0.05). Samples GL and GL-DH showed pasting temperatures that were significantly higher than cassava but lower than BECH (*p* < 0.05). Dry heating significantly increased the pasting temperature compared to the untreated GL. Paste viscosities (i.e., PV, HV and FV) were the highest for cassava starch, followed by sorghum flour, GL, BECH and GL-DH (*p* < 0.05). The paste viscosities of BECH were significantly lower than those of GL. Furthermore, dry heating of GL resulted in significant reductions in PV, HV and FV. BD was the largest for cassava starch and the lowest for sorghum, BECH and GL-DH. Sorghum flour showed the highest SB, followed by cassava starch, BECH and GL (*p* < 0.05). Dry heating significantly reduced SB compared to the untreated GL, which was in agreement with recent work on GL with similar treatments [31]. It was recently shown that changes in the pasting properties and water binding capacity of cowpea flour significantly affect the baking behaviour of bread dough made from sorghum, cowpea flour and cassava starch [31]. Therefore, differences in cowpea flour variety and concentration (in place of sorghum) were expected to modulate bread properties.

### 3.2. Baking Quality of Breads Made from Blends of Cassava, Sorghum and Cowpea Flours

Baking tests were performed to evaluate the effect of cowpea flour variety and level of addition on the baking performance of the dough (Table 4). Increasing GL content from about 9 to 27% in the dough resulted in an overall improvement in the baking performance, as indicated by the significant increase in SV, springiness and resilience and a concomitant decrease in crumb hardness. Despite the significant differences observed between the two cowpea varieties in terms of physicochemical properties (Table 3), replacing GL with BECH did not significantly affect bread properties (i.e., samples GL27 and BECH27, Table 4). Bread GL9-DH showed bread properties that were intermediate to GL9 and GL27, suggesting dry-heating may be a simple and effective technology to functionalise cowpea flour [31], although the size of the effect should be further optimized to justify the additional processing step.

Due to the lack of gluten, strain hardening is missing during fermentation and the early stages of baking with CRCs-based doughs. The mechanical resistance against gas pressure and collapse is merely provided by viscosity (or by shear modulus), as observed in cake batters [44]. From this standpoint, the differences in WBC among cassava, sorghum and cowpea flours are relevant as WBC will affect dough viscosity. A high level of cowpea flours in place of sorghum may contribute to improved dough stability during proofing and the early stages of baking, before starch gelatinization takes place, thus positively influencing bread volume.

During the baking stage with CRCs dough, fixation of the expanding foam structure is mainly related to the starch gelatinization process, due to the absence of the gluten thermo-setting mechanism. Starch gelatinization increases the viscosity and shear modulus of the batter dramatically and the swollen starch granules further support against normal forces in order to withstand collapse [45]. In bakery applications, such as cakes, where a gluten network is also absent, variations in the temperature at which starch gelatinization occurs largely determine the final volume [46]. Cowpea flours showed higher gelatinization temperatures than sorghum, which may have slightly prolonged the time available for further bubble expansion before structure setting. Additionally, the higher melting transition temperatures could also implicate a faster stabilisation of the starch paste during cooling. An optimum in viscosity is also likely to contribute to baking performance, with too high viscosity limiting expansion and too low resulting in collapse. As suggested by the RVA results, the paste viscosity was largely dominated by cassava starch. Raising the level of cowpea flour in place of sorghum or adding dry-heated cowpea flour most likely modulated paste viscosity, positively contributing to the increase in SV. Additionally, native and dry-heated cowpea flour have shown low paste breakdown due to the role of proteins around the starch granule acting as a physical barrier [31]. An increase in the amount of starch granules that maintain integrity and rigidity during baking can contribute to the continuity and strength of the starch phase upon cooling, thus resulting in improved springiness, cohesiveness and resilience [31].

### 3.3. Descriptive Sensory Profiling of Bread

Significant differences in appearance, odour and flavour attributes and in texture were observed among breads (Table 5). PCA analysis of the sensory profile also indicated that the first two principal components (i.e., PC_1_ and PC_2_) accounted for over 78% of the variance, thus providing a good representation of the differences among samples. Bread samples were differentiated in terms of texture (crumbliness, moistness) and yeasty flavour/fermented odour along the PC_1_, mainly dominated by samples GL27 and GL9-DH. PC_2_ was dominated by samples GL9 and BECH27, which were separated based on beany flavour, beany odour and darkness of the crumb (Figure 1A). The use of dry-heated cowpea (GL9-DH) resulted in a significant decrease in cereal odour and increased fermented odour compared to the other samples. Sample BENCH27 had the most intense beany odour and beany flavour while samples GL9 and GL9-DH, respectively, had the lowest (*p* < 0.05). Dry heating of cowpea provided bread GL9-DH with the most intense yeasty flavour. In a recent study, flatbreads made with the Bechuana variety of cowpea also showed a more intense beany odour and beany flavour compared to the Glenda variety [32]. Differences were attributed to the higher phenolic compounds in the more pigmented cowpea varieties [47,48]. Off-flavours are generated by the breakdown of polyunsaturated fatty acids by lipoxygenase to form secondary lipid oxidation products [49]. By binding to lipoxygenase present in the beans, phenolics limit the oxidation of unsaturated fatty acids during harvesting, thus controlling the beany aroma in legumes [50,51]. Flavour is one of the most important sensory attributes in terms of acceptability [52] and beany flavour considerably decreases the liking of bread enriched with legumes [53,54]. In this study, the addition of 27% cowpea flour enhanced the beany odour and flavour perception, particularly for BECH. The 9% addition of GL resulted in rather low scores for beany flavour (i.e., 2.17 on a 0-to-10-point scale), thus suggesting that such a level of inclusion may be suitable for bread applications. Thermal treatment has been reported to reduce beany flavour and increase yeasty flavour. Similar findings were recently reported for the thermal treatment of yellow pea [54]. However, thermal treatments can only partially attenuate the beany flavour [55]. The limited effects observed for sample GL9-DH may be explained by the low scores of beany flavour already attributed to sample GL9. The significant increase in yeasty and fermented flavour for sample GL9-DH may be a positive effect when associated with volatile compounds positively perceived by consumers [56] and should be further investigated in future studies.

BECH27 had the lightest crumb and GL27 had the darkest. In a recent study on flatbreads from composite flours, breads prepared with the Glenda variety were also perceived as darker than those with the Bechuana variety. Differences were attributed to the higher concentration of anthocyanins in the seed coats of the red variety [57]. A dark colour may negatively affect the liking of bread appearance by consumers [53]. From that perspective, the BECH variety may be preferred over the GL one.

Crumbliness is an important quality aspect of bread. Crumbliness is defined as the degree to which a sample fractures into pieces during mastication [58]. In our study, sample GL9-DH showed the highest crumbliness and lowest moistness, while sample GL27 showed the lowest crumbliness and highest moistness perception (*p* < 0.05). Perception of crumbliness and moistness were significantly correlated (R^2^ = 0.95, *p* < 0.05), with lower crumbliness scores observed with increasing moistness perception (Figure 1B). Crumbliness scores (i.e., both tactile and mouthfeel perception) were inversely related to instrumental cohesiveness and resilience (Figure 1B). However, the correlation was significant only between resilience and tactile crumbliness (R^2^ = 0.91, *p* < 0.05). For wet soft-solids, such as protein/polysaccharide gels, crumbliness perception increases with increasing recoverable energy [59], with recoverable energy and cohesiveness both being measures of the recovery after deformation [60]. Crumbly gels also exhibit low serum/water release [59]. Whether these principles also apply to breadcrumbs remains to be proven. In commercial gluten-free bread, the perception of crumbliness was inversely related to instrumental springiness [61]. Dryness perception of commercial wheat pan bread was inversely related to both instrumental cohesiveness and springiness [62]. On the contrary, no meaningful relations were observed between bread texture and texture profile analysis in studies conducted on gluten-free bread [63] and cassava composite bread [64]. Overall, a comparison of sensory and instrumental analysis on crumb properties from this study indicates that positive changes in mechanical properties, such as increased softness and cohesiveness/resilience, may be associated with positive sensory perception, i.e., less crumbliness.

Sandiness was perceived in all samples and changes in the composition of the flour blends and dry heating did not have significant effects. In preliminary trials, it was observed that both sorghum and cowpea flours contributed to sandiness. Pre-trials with sieving flours to below 200 µm or re-milling did not result in noticeable improvements (data not shown). For legumes, the hard-to-cook phenomenon is widely reported and attributed to structural and compositional changes induced by storage under hot and humid conditions, as encountered in many subtropical and tropical African countries [65]. Lignification of the cell wall contributes to increased cooking time [65] and could be one possible cause of sandiness perception in bread from this study, since the intact cowpea seeds were milled into flour. Additionally, an increase in starch crystallinity has been reported in beans during storage [66]. In cowpea flour, starch is tightly covered with protein material [67,68], which limits the water binding and swelling power of the starch [68]. From these data, it can be inferred that starch hydration during mixing and proofing may be a limiting factor during breadmaking, resulting in some crystalline starch material still being retained after baking. Sorghum starch can also contribute to this mechanism as starch granules are encapsulated by the hydrophobic kafirin [69], thus restricting the starch granules’ ability to absorb water. During cooking, this protein barrier is further strengthened by the formation of protein aggregates capable of resisting the combined dissociating action of urea and reducing agents, which further limits starch accessibility [70].

### 3.4. Consumers Evaluation of CRCs-Based Bread and Commercial Wholemeal Wheat Bread

The overall liking and sensory perception of CRCs-based bread and commercial wholemeal wheat bread were compared in the consumers’ test. The wholemeal wheat bread was chosen since the sorghum and cowpea flours were not refined. The CRCs-bread formulation for this study was formulation BENCH9 (Table 1); BECH was preferred to get a light crumb colour. The low quantity of cowpea flour was added to limit the beany odour and flavour. The CRCs bread received an average general liking of 5.0, indicating that the bread was neither liked nor disliked (Figure 2A). In particular, 47% of participants were in the group that liked the bread (scores 6–9), 12% did not like or disliked it (score = 5) and the remaining 41% did not like the bread (scores 1–4). The wheat bread received an average general liking of 6.84, meaning it was moderately liked. Differences in the overall liking of the two bread types were significant (*p* < 0.05).

With regards to the RATA tests, many significant differences were found for appearance, texture and taste attributes (Figure 2B–D). CRCs breadcrumbs were perceived as substantially darker and with a slightly more red tone compared to the predominant yellow tone and lighter appearance of the wheat bread. The CRCs bread had substantially smaller and more homogeneous pore size, providing a more dense appearance. The texture of CRCs breads was perceived by consumers as dry and crumbly compared to the wheat bread, which was soft, moist and chewy. Despite the fact that the wheat bread crumb was significantly softer, the hardness of the CRCs bread scored low in intensity. The CRCs bread was also perceived as sandy as compared to wheat bread, even though the latter contained bran particles. These results are in line with a recent evaluation of commercial gluten-free breads for which the texture sensations were described as hard, dry, crumbly and sandy [61]. Composite breads with sorghum were also reported to be sandy [71], which was attributed to the sorghum flour. Puerta and co-workers indicated that sandy and crumbly sensations in commercial gluten-free breads were correlated and could be both related to fragmentation of the bread crumb in the mouth as compared to wheat breads [61]. This mechanism may also contribute to the sandiness perception of the CRCs bread in this study. In our research, the sorghum and cowpea flours were eventually soaked in water for 1.5 h before use for breadmaking to minimize the perception of sandy-like particles. In preliminary evaluations, this approach seemed successful and substantially minimized sandiness perception.

The differences in flavour were related to salty, beany, tangy, sweet and sour attributes. However, the largest differences in intensity scores were limited to beany flavour and sour taste. Saltiness was more pronounced in the CRCs bread, likely due to a higher salt content than in commercial wheat bread. Even though differences in beany flavour were clearly perceived by consumers, this attribute received a low intensity score, suggesting that the chosen level of cowpea flour addition may also be suitable for white varieties such as Bechuana. The slight sourness and tanginess perceived in the CRCs bread may be considered a positive attribute, when not excessive, as they are naturally associated with sourdough breads [72].

### 3.5. Demonstration of Wheat Replacement in Real Life Conditions in SSA

One of the CRCs-based doughs in this study was evaluated for its implementation potential in SSA, as a means to reduce Africa’s dependency on imported wheat and valorise traditional and underutilised CRCs. Chapati-type flatbreads, the most popular convenience foods produced by street vendors [11], and tin breads, a popular breakfast and lunch staple in combination with condiments [73], were explored as relevant applications currently based on refined wheat.

Chapati were tested with street vendors in the Kasanvu Slum of Kampala (Uganda), by preparing popular dishes such as *Kikomando* (chapati served with beans; Figure 3F) and *Rolex* (a chapati baked with eggs and rolled together with vegetables; Figure 4D). Due to increasing wheat and fuel prices, local vendors are currently reducing the sizes of the chapati to maintain affordable prices (communication from vendors, June 2022). The flour mixture GL9 (Table 1) without yeast was provided to the street vendors. All other ingredients and utensils were supplied by the vendor. The CRCs-based dough could be prepared manually in a bowl and shaped into small dough balls as usually performed by the vendors (Figure 3A,B). The dough balls were first flattened by hand and then with a small roller (Figure 3C). The CRCs-based chapati was baked on an open-fire heated pan with oil, rolled-up and cut into pieces to be served with beans as ‘*Kikomando*’ (Figure 3D–F). The CRCs-based chapati was also heated together with eggs and then rolled with sliced tomatoes and served as a wrap named ‘*Rolex*’ (Figure 4). Vendors could prepare the CRCs-based chapati quite efficiently, which is one of the most important requirements for their laborious business, with customers waiting for freshly produced food on the go. The chapati were highly rollable, which was an important quality parameter for the street vendors for their versatile use.

Baking experiments for CRCs-based tin breads were also performed in Uganda in a BBROOD formal bakery in Tororo. The bakery produces and sells Western-type breads, including tin breads, which are becoming popular among the middle class in the evolving and rapidly growing urban market [11]. The dough mixing was performed with equipment available at the commercial bakery (Figure 5A). Tin breads were produced using the flour mixture GL9 (Table 1, Figure 5B). The baked breads were tasted at the BBROOD bakery shop in Muyanga (Kampala) by local customers. After tasting, about 20 people were informally asked for their feedback by the shop assistants. The customers’ (consumers) responses to the bread samples were observed and noted. Most people had a neutral to positive perception of the bread (i.e., 15 out of 20). Five out of the twenty indicated that the bread was too salty. Sandiness was mentioned by only one of the consumers. Breads using sorghum, cassava and cowpea locally sourced were also successfully produced using a recipe similar to GL9 Figure 5B), suggesting the adaptability to a local situation.

To date, wheat replacement initiatives have not been very successful, as consumers in many SSA countries perceive composite breads as a lower quality product and therefore prefer bread made from 100% wheat flour [29,74]. On the contrary, innovations that create new markets are critical to promote the use of CRCs ingredients [75]. The bread-type products in this study could be proposed and further studied as new products, thus overcoming the major pitfall for composite wheat breads not meeting consumers expectations on shape and appearance [29].

## 4. Conclusions

Cowpea flour proved to be functional in modulating the baking properties of the CRCs breads. High cowpea to sorghum ratio improved specific volume and instrumental texture by enhancing softness and cohesiveness. Substantial differences in WBC and pasting properties of cowpea flour compared to sorghum flour and cassava starch could be related to these improvements. Among varieties, the red Glenda cowpea was the most functional in reducing perceived crumbliness and increasing perceived moistness, probably due to its high WBC. Additionally, the Glenda variety scored better with regards to taste, as beany flavour was most pronounced in breads with high inclusion levels of the Bechuana variety. Dry-heating of cowpea flour substantially affected its functionality, but had limited effects on bread texture. The most significant effects were observed for the perceived yeasty and fermented flavour of the bread. Sensory tests with naive consumers indicated that the CRCs bread was the most distinct from commercial wheat bread for its crumbly texture and for the beany, sour and yeasty taste. Despite that, the CRCs bread was neither liked nor disliked, while the commercial bread was only moderately liked. These blends of CRCs could be successfully used in Uganda to produce chapati and tin breads with street vendors and local bakeries, respectively. Overall, the result of this study showed that blends of African local crops can be used to fully replace wheat in bread-type products. The selection of the cowpea to sorghum ratio and cowpea variety are important for modulating their texture and taste. These results are relevant to alleviate African dependency on imported wheat, thus potentially contributing to food and nutrition security in SSA in the current conditions of climate change and geopolitical crisis.

## Figures and Tables

**Figure 1 foods-12-00689-f001:**
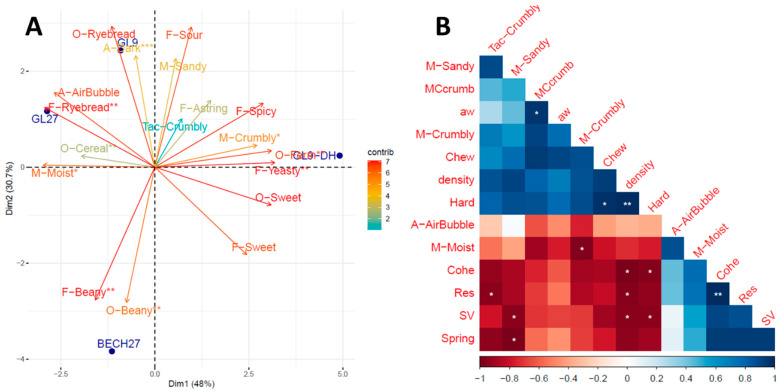
PCA biplot comparing the sensory attributes of the four CRCs based breads (**A**); correlation analysis between texture sensory attributes and instrumental properties of breads (**B**). Bread sample codes are explained in Table 1. Measured parameters: O-Cereal = cereal odour, O-sweet = sweet odour, O-Ryebread = rye bread odour, O-Beany = beany odour, O-Ferm = fermented odour, A-Dark = dark colour appearance, A-AirBubble = appearance air bubble size, Tac-Crumbly = crumbliness by hand (tactile), F-Sour = sourness, F-Ryebread = Rye bread flavour, F-Beany = beany flavour, F-Yeasty = yeasty flavour, F-Spicy = spicy flavour, F-sweetness = sweet taste, F-Astring = astringency, M-Sandy = sandy mouthfeel, M-Crumbly = crumbly mouthfeel, M-Moist = moistness in the mouth, SV = specific volume of bread, MCcrumb = % moisture content in the crumb, Chew = TPA chewiness, density = crumb density (g/mL), Hard = TPA hardness, Spring = TPA springiness, Cohe = TPA cohesiveness, Res = TPA resilience. For both PCA and correlation table: * *p* < 0.05, ** *p* < 0.01, *** *p* < 0.001.

**Figure 2 foods-12-00689-f002:**
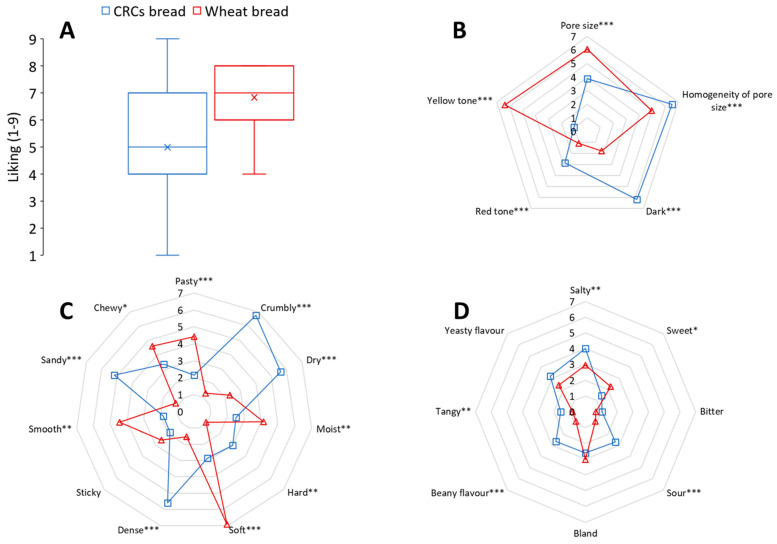
Overall acceptance of breads by consumers on a 1–9 scale (**A**) and comparison of the mean intensity scores for RATA test on a 0–9 point scale for attributes related to appearance (**B**), texture (**C**) and flavour (**D**). * *p* < 0.05, ** *p* < 0.01, *** *p* < 0.001.

**Figure 3 foods-12-00689-f003:**
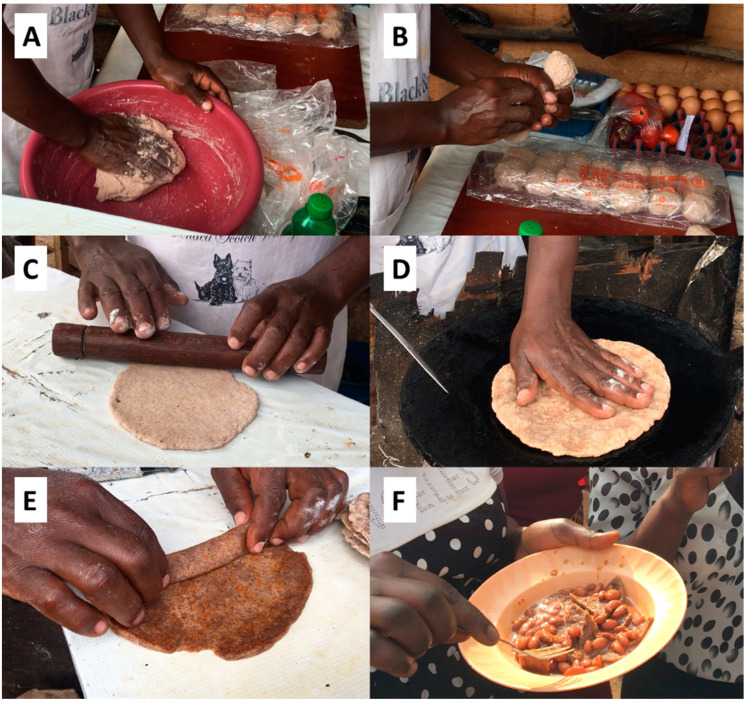
Main steps in the preparation of chapati by a street vendor using the CRCs-based dough GL9 (Table 1) without yeast. Dough mixing by hand in a plastic bowl (**A**), forming of dough balls by hand (**B**), sheeting with a roller (**C**), baking on a hot pan with additional oil (**D**), rolling the chapati before cutting (**E**) and serving Kikomando (chapati with beans) to the local consumers on the street (**F**).

**Figure 4 foods-12-00689-f004:**
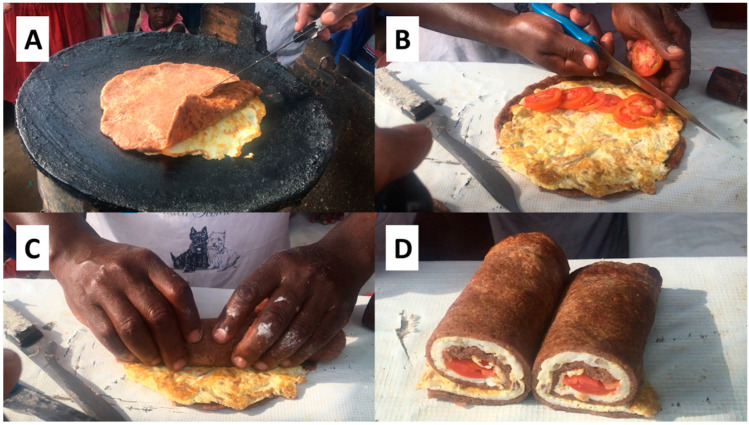
Preparation of a Rolex by the street vendor using the CRCs-based dough GL9 (Table 1) without yeast. After preparation and pre-baking as in Figure 3A–D, eggs are cooked on the hot plate and the chapati is added on top of the frying eggs (**A**). Once the eggs are cooked, the chapati is placed on a table and vegetables are added (**B**); the chapati is then rolled (**C**) and served to the local customers (**D**).

**Figure 5 foods-12-00689-f005:**
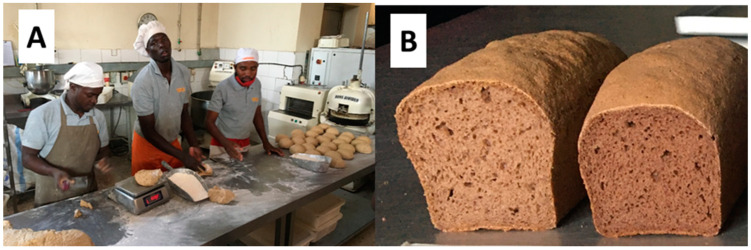
Bakers at BBROOD bakery in Tororo preparing the CRCs-based dough for tin bread baking (**A**); tin breads from the CRCs dough (formulation GL9 in Table 1) based on flours used in this study (**left**) and with flours locally sourced (**right**) (**B**).

**Table 1 foods-12-00689-t001:** Bread dough formulations tested for baking properties and descriptive sensory and for sensory with naive consumers.

Ingredients	Formulations in Baker’s %				
	GL9	GL27	BECH27	GL9-DH	BENCH9 **
Flours mixture *					
Sorghum	45.7	27.4	27.4	45.7	45.7
Cassava starch	45.7	45.7	45.7	45.7	45.7
GL	8.7	26.9			
BECH			26.9		8.7
GL-DH				8.7	
Salt	2.3	2.3	2.3	2.3	2.3
Dry yeast	5.0	5.0	5.0	5.0	5.0
Rapeseed oil	3.7	3.7	3.7	3.7	3.7
Sucrose	3.7	3.7	3.7	3.7	3.7
Psyllium flour	7.3	7.3	7.3	7.3	7.3
Water	108.7	108.7	108.7	108.7	108.7

* Sorghum, cassava and cowpea flours account together for the total flour; the remaining ingredients are expressed as percentage of the flour mixture. GL = Cowpea flour from Glenda variety; BECH = cowpea from Bechuana variety; GL-DH = dry heated cowpea flour Glenda variety. ** Formulation used only for sensory evaluation with naive consumers.

**Table 2 foods-12-00689-t002:** List of descriptors and definitions used in the RATA test with naive consumers.

Attribute	Definition
**Appearance**	
Pore size	Size of holes inside a loaf
Homogeneity of pores	Observation of regular size of pores
Colour: dark	Perception of dark colour
Colour: red tone	Perception of red colour tones
Colour: yellow tone	Perception of yellow colour tones
**Texture**	
Hard	Related to the force required to bite
Soft	Related to the force required to bite
Dense	Tightly packed crumb structure, more closed crumb structure
Sticky	Adhering or sticking to oral cavity
Smooth	Degree of perceived smoothness of bread
Sandy	Sensation that describes presence of particles in oral cavity
Chewy	Related to the number of chews required before swallowing
Pasty	Sensation that describes the formation of a dough of the bolus
Crumbly	Easily breaking into small fragments
Dry	Degree of drying effect, amount of saliva absorbed by the sample
Moist	Amount of moisture perceived of the product
**Taste**	
Salty	Perception of salt
Sweet	Perception of sugar taste
Bitter	Perception of bitter taste
Sour	Perception of sour taste
Beany flavour	Having a flavour associated with cooked dry beans
Bland	Lacking taste
Tangy	Having a strong piquant flavour.
Yeasty flavour	Having a flavour associated with (dry) yeast

**Table 3 foods-12-00689-t003:** Physicochemical properties of the flours used in the study.

	Cassava	Sorghum	BECH	GL	GL-DH
*WBC (g/g)*	1.5 ± 0.1 a	2.1 ± 0.2 b	3.1 ± 0.2 c	3.7 ± 0.3 d	3.1 ± 0.2 cd
*DSC parameters*					
Tonset (°C)	60.1 ± 0.2 a	68.0 ± 0.2 d	67.0 ± 0.6 cd	64.6 ± 1.0 b	66.1 ± 0.5 bc
Tpeak1 (starch) (°C)	71.1 ± 0.1 a	74.4 ± 0.1 b	78.5 ± 0.1 d	76.0 ± 0.2 c	76.0 ± 0.1 c
Tpeak2 (protein) (°C)	-	-	87.4 ± 1.1	88.3 ± 0.6	87.7 ± 0.9
$\Delta H_{}$ (kJ/mol)	17.0 ± 0.5 b	14.9 ± 1.6 b	8.9 ± 1.2 a	7.8 ± 0.5 a	8.6 ± 0.6 a
*RVA parameters*					
PT (°C)	70.7 ± 0.0 a	92.2 ± 0.1 e	86.6 ± 0.1 d	80.3 ± 0.1 b	82.8 ± 0.2 c
PV (cP)	2397 ± 4 e	487 ± 1 d	219 ± 3 b	277 ± 15 c	130.5 ± 1 a
HV (cP)	1094 ± 2 d	484 ± 1 c	216 ± 2 b	247 ± 21 b	129 ± 1 a
BD (cP)	1303 ± 6 c	4 ± 1 a	4 ± 1 a	30 ± 6 b	1.5 ± 1 a
FV (cP)	1471 ± 2 d	967 ± 4 c	363 ± 5 b	342 ± 13 b	174.5 ± 2 a
SB (cP)	377 ± 0 d	483 ± 2 e	147 ± 3 c	96 ± 8 b	45.5 ± 1 a

Values with different letters within each row indicated significant differences in ANOVA analysis (*p* < 0.05). GL = Cowpea flour from Glenda variety; BECH = cowpea from Bechuana variety; GL-DH = dry heated cowpea flour Glenda variety.

**Table 4 foods-12-00689-t004:** Specific volume (SV) and crumb properties of the CRCs based breads.

	GL9	GL27	BECH27	GL9-DH
SV (mL/g)	1.73 ± 0.03 a	1.82 ± 0.02 bc	1.84 ± 0.00 c	1.77 ± 0.03 ab
*Crumb properties*				
Moisture (%)	52.5 ± 0.1 a	52.4 ± 0.0 a	52.4 ± 0.1 a	52.5 ± 0.1 a
Hardness (N)	21.1 ± 1.7 b	16.2 ± 1.6 a	16.4 ± 1.8 a	20.6 ± 1.3 b
Springiness	0.887 ± 0.012 a	0.917 ± 0.018 b	0.915 ± 0.009 b	0.904 ± 0.010 b
Cohesiveness	0.423 ± 0.017 a	0.477 ± 0.012 b	0.462 ± 0.016 b	0.436 ± 0.008 a
Resilience	0.191 ± 0.011 a	0.225 ± 0.008 c	0.213 ± 0.010 bc	0.201 ± 0.005 ab

Values with different letters within each row indicated significant differences in ANOVA analysis (*p* < 0.05). GL9 = Bread with 9% Glenda cowpea flour; GL27 = Bread with 27% Glenda cowpea flour; BECH27 = Bread with 27% Bechuana cowpea flour; GL9-DH = Bread with 9% dry-heated Glenda cowpea flour.

**Table 5 foods-12-00689-t005:** Sensory profile analysis of the different breads: mean scores and Tukey’s post hoc groups.

Attribute	GL9	GL27	BECH27	GL9-DH	*p*-Values
*Appearance*					
Darkness	4.74 bc	6.28 a	3.41 c	4.94 ab	<0.001
Pore size	3.91	4.17	3.64	3.46	0.232
*Odour*					
Cereal odour	4.89 a	3.51 ab	4.20 ab	2.78 b	0.004
Sweet odour	2.52	2.24	2.84	3.70	0.056
Ryebread odour	3.79	3.86	2.92	3.3	0.452
Beany odour	1.67 b	3.06 ab	3.89 a	2.54 ab	0.009
Fermented odour	1.81 b	1.59 b	1.54 b	5.46 a	<0.001
*Flavour/Taste*					
Ryebread flavour	3.32 a	3.45 a	2.69 ab	1.91 b	0.007
Beany flavour	2.17 b	2.70 ab	4.08 a	1.94 b	0.004
Yeasty flavour	1.88 b	1.18 b	1.73 b	4.22 a	0.001
Spicy flavour	2.12	1.84	1.56	2.94	0.122
Sourness	3.94	3.95	3.06	4.03	0.265
Sweetness	1.96	2.16	2.54	2.82	0.353
Astringency	2.18	2.56	2.06	2.65	0.390
*Texture*					
Crumbliness tactile	6.17	5.55	5.79	5.84	0.774
Sandiness	3.64	3.06	3.01	3.26	0.642
Crumbliness mouthfeel	5.61 ab	4.56 b	5.13 ab	5.95 a	0.019
Moistness	2.99 ab	3.58 a	3.13 ab	2.55 b	0.049

Attribute intensities scored with a 0–10 continuous line scale with 0 = non-perceivable and 10 = very intense. Values with different letters within each row indicated significant differences in ANOVA post hoc analysis (*p* < 0.05). GL9 = Bread with 9% Glenda cowpea flour; GL27 = Bread with 27% Glenda cowpea flour; BECH27 = Bread with 27% Bechuana cowpea flour; GL9-DH = Bread with 9% dry-heated Glenda cowpea flour.

## Data Availability

Data available upon request to corresponding author.

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
