# Peer review of "Bread Products from Blends of African Climate Resilient Crops: Baking Quality, Sensory Profile and Consumers’ Perception"

_foods, 2023, doi:10.3390/foods12040689_

Round 1
Reviewer 1 Report
The manuscript entitled "Bread products from blends of sorghum, cowpea and cassava flour to promote healthy diets and food security: baking quality, sensory profile and consumers' perception" by Renzetti, Aisala, Ngadze, Linnemann and Noort presents a research on the use of crops grown in the sub-Saharan region for the production of baked foods of regional importance, with the aim of reducing the dependence of these countries on imported crops, such as wheat. The work has a clear objective, the experimental design is adequate and the results are well presented and discussed.
Some concerns need to be resolved:
Ln 117-118. It is not clearly indicated why the authors performed heat treatment only on one of the varieties (GL). I believe that the reason for this choice should be made explicit.
Ln 124. Since the work was carried out at different stages and at different locations, I suggest specifying a little more "locally sourced".
Ln 135- "pellet was drained for 15 min at an angle of 45°, 134 dried..." what temperature and time were used to dry the sample?
2.2.1 is described in some detail, while 2.2.2 and 2.2.3 only cite the methodology. I suggest that they briefly describe the procedures used.
Ln 152. Please, check this error along the manuscript.
Table 1. Only one of the varieties was evaluated at two doses (9% and 27%). First, the reason for the choice should be made explicit. Second, in the abstract, authors mention that "Increasing cowpea flour addition from 9 to 27% significantly improved bread volume...", this generalization may be misleading.
Ln 186-189. The choice of panelists from Finland, when it is stated at the outset that the products are focused on African countries, is striking. Is there any justification for this choice other than convenience?
Ln 186-189. This information seems to be repeated below, please corroborate.
Ln 244. "but using BECH instead of GL". This is confusing. The formulations evaluated (Table 1) do not include BECH9. I have the impression that the experimental design should be better ordered so as not to confuse the reader.
Ln 278. "The WBC of dry-heated GL flour was intermediate to GL and BECH". Perhaps a brief justification of this result can be added. Do the authors believe that this result may be due to a change in protein conformation, and/or starch crystallinity, for example?
Table 3. Please, add the reference for the abbreviations.
Ln 321-322. "confirming dry-heating as a simple and effective technology to functionalize cowpea flour". Although an increase in the specific volume is observed, it should be evaluated whether this increase (1.73 vs. 1.77) justifies the time and energy input for the process.
Table 5. Why are there significant differences only for some parameters?
Figure 1. Please check this figure, the text in the biplot is overlapped.
Ln 501. Chapatti
Author Response
The manuscript entitled "Bread products from blends of sorghum, cowpea and cassava flour to promote healthy diets and food security: baking quality, sensory profile and consumers' perception" by Renzetti, Aisala, Ngadze, Linnemann and Noort presents a research on the use of crops grown in the sub-Saharan region for the production of baked foods of regional importance, with the aim of reducing the dependence of these countries on imported crops, such as wheat. The work has a clear objective, the experimental design is adequate and the results are well presented and discussed.
We thank the reviewer for his interest in our work. We have addressed the reviewer’s comment to improve the quality of the manuscript
Some concerns need to be resolved:
Ln 117-118. It is not clearly indicated why the authors performed heat treatment only on one of the varieties (GL). I believe that the reason for this choice should be made explicit. In a previous study, the Glenda variety was extensively studied under different dry-heating conditions, but its contribution to the sensory profile of bread was not evaluated. Authors have revised the section to provide further explanation for the choice
Ln 124. Since the work was carried out at different stages and at different locations, I suggest specifying a little more "locally sourced".
The sentence has been revised accordingly to specify
Ln 135- "pellet was drained for 15 min at an angle of 45°, 134 dried..." what temperature and time were used to dry the sample? We have added the information accordingly
2.2.1 is described in some detail, while 2.2.2 and 2.2.3 only cite the methodology. I suggest that they briefly describe the procedures used. We have added further information as suggested
Ln 152. Please, check this error along the manuscript. There seems to be an error when the word file we submit is converted by the journal website. We will submit a pdf file directly. This should be now resolved
Table 1. Only one of the varieties was evaluated at two doses (9% and 27%). First, the reason for the choice should be made explicit. Second, in the abstract, authors mention that "Increasing cowpea flour addition from 9 to 27% significantly improved bread volume...", this generalization may be misleading. The main reason for the limited variations was by limited resources of materials from the same batch. Hence the comparison between varieties was chosen at highest level to maximize differences. We have revised the section to provide further clarifications. We agree with the misleading message and revised by specifying the variety used.
Ln 186-189. The choice of panelists from Finland, when it is stated at the outset that the products are focused on African countries, is striking. Is there any justification for this choice other than convenience? The purpose of the sensory profiling was to detect relative differences in attribute intensities between the tested products rather than hedonic responses. From this perspective, the trained panelists are used as an “instrument” and the cultural background of the background has less impact than in liking tests. It was most important to have a good and well-trained panel which was available for the study. In our case, this was at VTT.
Ln 186-189. This information seems to be repeated below, please corroborate. There was an error. The section has been revised
Ln 244. "but using BECH instead of GL". This is confusing. The formulations evaluated (Table 1) do not include BECH9. I have the impression that the experimental design should be better ordered so as not to confuse the reader. We have revised the table and the text to further clarify
Ln 278. "The WBC of dry-heated GL flour was intermediate to GL and BECH". Perhaps a brief justification of this result can be added. Do the authors believe that this result may be due to a change in protein conformation, and/or starch crystallinity, for example? In a recent study on Glenda cowpea flour, we have shown that WBC decreases with increased severity of the treatment while no significant changes are observed on starch gelatinization. At same time, soluble solids significantly decrease (i.e. mostly soluble proteins). In agreement with previous studies, we suggest that both increased protein hydrophobicity and starch annealing can contribute to the observed effect. We have added a sentence and references to clarify.
Table 3. Please, add the reference for the abbreviations. Reference has been added
Ln 321-322. "confirming dry-heating as a simple and effective technology to functionalize cowpea flour". Although an increase in the specific volume is observed, it should be evaluated whether this increase (1.73 vs. 1.77) justifies the time and energy input for the process. We have revised the sentence to provide a more balanced consideration.
Table 5. Why are there significant differences only for some parameters? We have revised the table to further clarify, showing also the p values for attributes with no significant differences
Figure 1. Please check this figure, the text in the biplot is overlapped. The plot is automatically generated in R with coding to avoid overlapping. This occurs only for GL9-DH and O-Ferm labels. We could not solve the issue, unless we split the graph in loading plot and score plot. However, the number of samples is limited and we believe that by looking at the codes provided in the footnote the labels GL9-DH and O-Ferm can be easily recognized. The increase in fermented odour with samples GL9-DH is also indicated in the text. Hence, we believe the message is clear.
Ln 501. Chapatti. We have revised in the text to chapati
Reviewer 2 Report
I recommend minor revisions.
General comment
The topic is worth investigating. The experimental design is simple, yet accurate and scientifically sound. Results are well presented.
Two areas must be improved heavily:
- Scientific discussion of sections "3.2" and "3.3”.
- In-text citations (multiple errors present, possibly due to EndNote.
Specific.
Specific comments
Abstract: add key values for instrumental analysis.
Add a space among sections and sub-sections
Line 120: indicate the variety of cassava used if known
Lines 138-144: indicate the DSC parameters flow rate and temperature range
Line 186: Where is the title of section 2.2.6?
Line 221: DO you mean “crumliness” or “crumbliness”?
Line 245 and others: please fix the citation error “Error! Reference source not found”
Section 3.1, Line 288: Show at least 1 representative thermogram for GL and BECH depicting the two endothermic transitions. Why was only one transition integrated in the table? Explain
Lines 325-351: Please discuss papers that considered the effect of particle size on starch gelatinization, using instruments such as Microscopy/Image Analysis (structure) and the Mastersizer (particle size)
Line 369: Remove “the” before phenolics”
Line 429: remove the extra space between “trials” and “it”
Figure 2: Figure and legend in the same page
Line 479-483: Have you measured the granulometry (sieves) or the particle size (Mastersizer)?
Lines 577-579: What fermentation type are you recommending: yeast based, sourdough by spontaneous bacteria and yeasts (wild), sourdough by selected bacteria (Lactobacillus)?

Author Response
Responses in attached file

Reviewer 3 Report
Q: What is the novelty of this study?
Q: Why did the authors investigate just the effects of cowpea flour? How about the effects of sorghum and cassava?
Q: There are a lot of errors in the manuscripts (please refer to the phrases ‘Error! Reference source not found’ in the manuscript). Please amend the errors.
Q: Please delete Figures 3, 4 and 5.
Q: Table 5: How to conduct the statistical analysis? Please exhibit the statistical results properly.
Author Response
Responses in attached file

Reviewer 4 Report
Thanks to the authors of the manuscript for the interesting research!
However, first it is necessary to arrange technical matters in the manuscript:
1. References to scientific articles in the text (Error! Reference source not found.) – for example, Lines 151-152; 244; 249; 276; 286; 318-319; 355; 359 etc. I don't know where the problem is, but there are many such errors in the document submitted to the system.
2. Please revise the manuscript and reduce abbreviations where possible. For example, Table 3; Lines 295-306 or 355-366. My recommendation – if any abbreviations are used in the Table, then give their transcripts after the Table, so that the reader can understand the text more easily.
3. In the manuscript, it is necessary to use the correct script of cassava, sorghum and cow pea varieties - "Glenda" or 'Glenda' or Glenda'. Please use one of them!
4. Please review whether the correct designation for the type of bread in English is used - tin bread=thin bread. Is it a special designation for a type of bread? dough fermentation cabinet=dough fermentation chamber. Table 1 – Baker's= Samples or Sample codes. Why are Table 1 Samples in %?
5. Acknowledgments Line 593 check for correctness - Dick van Breggen van der Breggen.
6. Incorrect reference entry, which does not give the opportunity to fully follow the references in the manuscript used.
It is necessary to clarify the title of the manuscript, because the manuscript is based on the sensory properties and profile of the bread and not on their health and food safety. My recommendation – Bread products from blends of sorghum, cowpea and cassava flour: baking quality and sensory profile. Therefore, do not emphasize in the manuscript health and food safety of products.
It is not clear what the authors of the manuscript wanted to say with subsection 3.5. Opportunities for implementation of results in SSA for food and nutrition security. Most of the description (Lines 500-518 and 525-536) are the methodology for making bread and pictures of bread making in SSA - what does all this have to do with nutrition security? Only 20 consumers participated in the evaluation of CRC-s based thin bread. I believe that this is too small a number of consumers if the sensory evaluation was done in the SSA on the street. Also, it is not clear what method was used in the sensory evaluation of the samples. I believe that this subsection is redundant or it need significant improvements.
2.1. Materials (Lines 108-118) – It is recommendation to give this data in a Table, then the information is easier to perceive and understand, then the codes of samples used will be more visible.
Line 136 – check if everything is in order with the number of brackets used in the formula.
Lines 186-190 - it is not clear what the authors wanted to say with this paragraph - the information is repeated.
Line 211 – air bubble size= pore size (that's the term you used afterwards); crmbly=crumbly
Table 2 – what is the explanation and what is the difference between dry and moist attributes and their definition twice? The descriptions are the same.
Line 311 – mixtures=blend
Table 5 – what method was used? If this is not stated, then the following descriptions and explanations are not clear either.
Conclusions
Lines 559-561 - this is not a conclusion, but a finding. Please remove or modify.
Lines 577-579 - do not really come from this study.
The other conclusions are also too broad and general. These are not conclusions, but more of a summary. Please specify!
Author Response
Responses in attached file

Reviewer 5 Report
The manuscript is interesting approach towards replacement of wheat in SSA, related to Russia-Ukraine war, wheat shortage and global inflation. From scientific point of view it was prepared correctly. Unfortunately it is not prepared thoroughly form editorial point of view.
Detailed remarks:
Abstract is my opinion too long and has little information about results.
Line 24 – melting?
Line 62 – citation in wrong format
Line 146-149 please provide concentration
Line 152 – reference issue
Table 1 is not clear with regards to units and abbreviations
Line 186 title issue
Line188 VTT?
Line 191-193 repetition
Line 213-218 repetition
Line 244 reference error
Line 249 as above
Table 2 dry and moist are duplicated in texture section
Line 276 reference error
Table 3 RVA data is lacking units (temperature and viscosity), moreover please add explanatory notes for statistics
Line 285 style and reference error
Line 313 reference error
Line 318 as above
Line 319 as above
Line 328 the reference in wrong format
Line 355 reference error
Line 357 principal component 1 was not explained earlier
Line 359 reference error
Table 5 please inform about scale (table should be self explanatory)
Line 398, 400, 418, 451,455, 465, 501, 503, 505, 509, 510, 512, 514, 521, 529, 530, 532, 538, 541, 542, reference error
Line 505 please provide n value
Section 3.5 lacks scientific approach
Author Response
Responses in attached file

Round 2
Reviewer 1 Report
The authors responded to all the concerns raised. (Please, remember that highlighting or clearly indicating in which lines the changes were made greatly expedites the review process).
Reviewer 3 Report
The authors have addressed my concerns.
Reviewer 4 Report
Thanks!
Reviewer 5 Report
Thank you for revision of the manuscript, I don’t have any further comments.